# Effects of a Low-Carbohydrate High-Fat Diet Combined with High-Intensity Interval Training on Body Composition and Maximal Oxygen Uptake: A Systematic Review and Meta-Analysis

**DOI:** 10.3390/ijerph182010740

**Published:** 2021-10-13

**Authors:** Jing Hu, Zhen Wang, Bingkai Lei, Junping Li, Ruiyuan Wang

**Affiliations:** School of Sports Science, Beijing Sport University, Beijing 100084, China; 2019210193@bsu.edu.cn (J.H.); wz4711@126.com (Z.W.); leibingkai0520@163.com (B.L.)

**Keywords:** low-carbohydrate high-fat diet, high-intensity interval training, body composition, maximal oxygen uptake, weight loss, meta-analysis

## Abstract

The low-carbohydrate high-fat (LCHF) diet has recently been subject to attention on account of its reported influences on body composition and physical performance. However, the combined effect of LCHF with high-intensity interval training (HIIT) is unclear. A systematic review and meta-analysis were conducted to explore the effect of the LCHF diet combined with HIIT on human body composition (i.e., body weight (BM), body mass index (BMI), fat mass (FM), body fat percentage (BFP), fat-free mass (FFM)) and maximal oxygen uptake (VO_2max_). Online libraries (PubMed, Web of Science, EMBASE, Cochrane Library, EBSCO, CNKI, Wan Fang) were used to search initial studies until July 2021, from which 10 out of 2440 studies were included. WMD served as the effect size with a confidence interval value of 95%. The results of meta-analysis showed a significant reduction in BM (WMD = −5.299; 95% CI: −7.223, −3.376, *p* = 0.000), BMI (WMD = −1.150; 95% CI: −2.225, −0.075, *p* = 0.036), BFP (WMD = −2.787; 95% CI: −4.738, −0.835, *p* = 0.005) and a significant increase in VO_2max_ (WMD = 3.311; 95% CI: 1.705, 4.918, *p* = 0.000), while FM (WMD = −2.221; 95% CI: −4.582, 0.139, *p* = 0.065) and FFM (WMD = 0.487; 95% CI: −3.512, 4.469, *p* = 0.814) remained unchanged. In conclusion, the LCHF diet combined with HIIT can reduce weight and fat effectively. This combination is sufficient to prevent muscle mass loss during LCHF, and further enhance VO_2max_. Further research might be required to clarify the effect of other types of exercise on body composition and physical performance during LCHF.

## 1. Introduction

Body composition is influenced by nutritional and physical activity intervention. The low-carbohydrate high-fat (LCHF) diet is a re-emerged dietary approach, which is characterized by decreased carbohydrate intake (approximately 50 g/d) and high levels of fat consumption with adequate protein provided [1]. The global prevalence of overweight and obesity has multiplied in recent decades [2]. Obesity is a complex disease and a major risk factor for cardiovascular and metabolic disorders, especially atherosclerosis, type II diabetes, and metabolic syndrome [3], and it increases the risk of death [4]. LCHF has become fashionable because of its potential to induce rapid weight loss, including ketogenic diets (KDs), the Zone diet, the South Beach diet, the Atkins diet, and other carbohydrate-restricted diets (CRD) [5,6]. The purpose of this low-carb, high-fat dietary strategy of LCHF is to increase the utilization of fat as muscle fuel and to keep the body at high levels of circulating ketones for weight loss [7,8]. Some studies revealed that individuals who consumed the LCHF diet experienced greater body weight (BM) and fat mass (FM) loss than those adhering to other dietary interventions, such as a calorie-restricted diet or low-fat diet [9,10,11] [9(RCT),10(RCT).11(RCT)], which highlights the effectiveness of LCHF in managing obesity. In addition to being a dietary solution for obese people, LCHF is also effective in the treatment of type 2 diabetes [12], epilepsy [13], Alzheimer’s disease [14], Parkinson’s disease [14], and in reducing the risk of asthma [15]. In addition, LCHF is also used by exercise enthusiasts to enhance aerobic capacity [16] (e.g., maximum oxygen uptake).

Although LCHF has gained considerable attention in the dietary treatment of chronic diseases such as obesity and type 2 diabetes [12], there is much controversy. LCHF not only may reduce lean mass (LM) and free fat mass (FFM) [17,18,19,20,21] [17 (RCT), 18 (RCT), 19 (NRCT), 20 (NRCT), 21 (RCT)] but also may cause several adverse reactions, including constipation, bad breath, muscle cramps, headache, diarrhea, weakness, and skin rashes [22]. Furthermore, LCHF may lead to liver inflammation, liver fat accumulation, insulin resistance, dyslipidemia, and hypertension, increasing the risk of cardiovascular disease [23]. However, unlike these dietary approaches, exercise interventions, such as high-intensity interval training (HIIT) or resistance training (RT), are known to bring beneficial changes in body function and composition, which imply the combination with exercise may counteract this negative effect. It is now well established from various studies that HIIT or RT improve physical function (e.g., ameliorate insulin resistance [24,25], reduce liver fat accumulation [26,27], decrease cardiovascular disease risk [28,29], improve aerobic capacity [30,31], enhance muscle strength [32], and increase LM and FFM [21,30,33,34] [21 (RCT), 30 (RCT), 33 (RCT), 34 (RCT)]. The benefits of HIIT and RT on LM and FFM are attributed to increased energy expenditure due to muscle contraction, which enhances adipose metabolism in muscle tissue and muscle growth-related factor gene expression [35]. Nevertheless, a recent meta-analysis presented that individuals assigned to a ketogenic diet showed fat-free mass loss, and this was not ameliorated by the combination of resistance training [36]. Thus, whether other exercise forms, such as HIIT, could reverse this negative effect requires attention.

In addition to the anti-obesity effect of LCHF, it has also become a popular nutritional strategy for athletes. Carbohydrates, present in muscles and liver as glycogen, have been confirmed as the primary fuel source for high-intensity exercise, and they are important for maintaining long-lasting exercise performance [37]. Traditional exercise nutrition guidelines propose high levels of carbohydrate consumption to increase skeletal muscle and liver glycogen content, thereby improving endurance performance [38,39]. However, the body’s glycogen storage capacity is limited (approximately 100 g in the liver and 300–700 g in the muscle) [40], which may lead to glycogen depletion during prolonged exercise, causing body fatigue. Therefore, the dietary approach (i.e., LCHF), which could increase endogenous fat oxidation and reduce the body’s dependence on glycogen during prolonged exercise [8], is receiving much attention. Dostal et al. show a greater increase in total time to exhaustion in the LCHF diet group than those in the habitual diet group [41]. However, the positive of LCHF on physical performance is debated. Several lines of evidence suggest that maximal oxygen uptake (VO_2max_) or peak oxygen uptake (VO_2peak_) were unchanged or decreased after consuming an LCHF diet [16,42,43,44]. HIIT, on the other hand, has also been proven to promote physical performance. Data from several studies verified that HIIT improves VO_2max_ among healthy and overweight/obese adults [44,45,46] [45 (RCT), 46 (RCT)]. However, the combined effect of LCHF and HIIT on VO_2max_ is still unknown.

Therefore, this study intended to conduct a systematic review and meta-analysis to explore the effect of the LCHF diet combined with HIIT on human body composition (i.e., BM, body mass index (BMI), FM, body fat percentage (BFP), and FFM) and VO_2max_. We hypothesized that LCHF combined with HIIT would be effective in improving body composition and aerobic capacity. To our knowledge, this is the first systematic meta-analysis to investigate the combined effects of LCHF and HIIT on human body composition and aerobic capacity.

## 2. Materials and Methods

### 2.1. Literature Search

Literature search, study selection, data extraction, and data analysis were performed according to PRISMA (Priority Reporting Entries for Reviews and Meta-Analyses). To find relevant prospective studies, a systematic literature search was conducted in electronic databases, including PubMed, Web of Science, EMBASE, Cochrane Library, EBSCO, CNKI, and Wan Fang, from build to July 2021. Our search strategy included the following keywords: “ketogenic diet”, “KDs”, “low carbohydrate diet”, “keto-adaptation”, “carbohydrate-restricted diet”, “high intensity interval training”, “sprint interval training”, “intermittent training”, and “aerobic interval training HIIT”. In addition, we reviewed the reference lists of previous systematic reviews and meta-analyses in the area.

### 2.2. Inclusion and Exclusion Criteria

Studies were considered eligible for inclusion if they met all of the following criteria: (a) study in adults (≥18 years); (b) dietary intervention was a low-carbohydrate high-fat diet (carbohydrate intake <50 g/d); (c) exercise intervention was high-intensity intermittent exercise; (d) duration of intervention was more than two weeks; (e) included outcomes: BM, BMI, FFM, BFP, FM, and VO_2max_; and (f) trial design was a crossover or parallel randomized controlled trial (RCT). Exclusion criteria were: (a) not meeting inclusion criteria; (b) combined with other types of dietary interventions; (c) full text not available; and (d) animal, review, and experimental studies.

### 2.3. Quality Assessments

Study quality was assessed using a “risk of bias” approach, as recommended by the Cochrane risk of bias tool [47]. This method classifies bias in randomized studies as “low”, “high” or “unclear” based on the presence of seven processes (random sequence generation, allocation concealment, blinding of participants and personnel, blinding of outcome assessment, incomplete outcome data, selective reporting, and other biases).

### 2.4. Data Extraction

Literature screening and data extraction were performed independently by two researchers (JH and ZW), and a third researcher (LB) assisted in resolving disagreements when they occurred. The following data were extracted from the included studies: (a) type of study; (b) population; (c) age of the population; (d) type of intervention; (e) duration of intervention; (f) measurement tools; and (g) outcome. Outcomes were meta-analyzed using the mean and standard deviation changes between the baseline and final values for each outcome, converting median values to means and 1st–3rd quartiles to standard deviations, respectively.

### 2.5. Statistical Analysis

Meta-analysis was completed using stata15.0 (College Station, TX, USA). WMD was used as the effect size with a confidence interval value of 95%. Q test with I^2^ test was used to investigate whether there was heterogeneity among the studies, and a fixed-effects model was applied because *p* > 0.10 and I^2^ < 50% indicated better homogeneity between studies. To test for possible bias in the included studies, their funnel plot symmetry as well as the Begg’s rank correlation test were examined.

## 3. Results

### 3.1. Study Selection

A total of 2440 articles were yielded through online libraries, and 433 duplicate articles were removed. After accessing the title and abstract, 1929 studies were excluded for the following reasons: unrelated title and abstract (1432), animal studies (375), and review studies (122). A total of 78 potentially related articles were screened by full text, while 68 articles were excluded owing to the absence of LCHF or/and HIIT interventions. Overall, 10 studies were included. The flow diagram of study selection is shown in Figure 1.

### 3.2. Characteristics of the Included Studies

The characteristics of the included studies are summarized in Table 1. Among them, seven studies were designed as RCT [39,40,41,42,43,44], and three studies were designed as non-RCT [33,35,46]. The subject’s ages ranged from 18 to 60 yr. Two studies contained only male subjects [48,49], three enrolled only female participants [43,50,51], and five studies included both sexes [41,52,53,54,55]. The participants included obese males and/or females [43,50,51,53,54], endurance-trained males [49], untrained healthy males and/or females [41,48], and metabolic syndrome males and females [52,55]. The intervention duration ranged from 2 weeks to 14 weeks [17,56,57,58].

### 3.3. Results from Quality Assessments

Risk of bias (ROB) was used to assess the risk of bias. The ROB of random sequence generation was “low” in seven and “high” in three studies. Regarding allocation concealment, the ROB was “low” in ten studies. For the blinding of the participants and personnel, the ROB was “high” in three, and “unclear” in seven studies. The ROB of blinding of the outcome assessments was “low” in six and “unclear” in four studies. For incomplete outcome data, the ROB was “low” in ten studies. Overall, all studies met the quality level. Therefore, no studies were excluded (Figure 2 and Figure 3).

### 3.4. Publication Bias

Begg’s test confirmed the absence of publication bias for studies assessing the effect of LCHF on BM (*p* = 0.474), BMI (*p* = 0.462), FM (*p* = 1), BFP (*p* = 0.462), FFM (*p* = 0.308), and VO_2max_ (*p* = 0.452). Furthermore, funnel plots exhibited a symmetric distribution, which also proved this point (Appendix A).

### 3.5. Results for Body Components and Maximal Oxygen Uptake

The results of meta-analysis indicate that participants that adhered to the LCHF diet and HIIT showed a significant reduction in BM (WMD = −5.299; 95% CI: −7.223, −3.376, *p* = 0.000, Figure 4), BMI (WMD = −1.150; 95% CI: −2.225, –0.075, *p* = 0.036, Figure 5), and BFP (WMD = −2.787; 95% CI: −4.738, −0.835, *p* = 0.005, Figure 6), and a significant increase in VO_2max_ (WMD = 3.311; 95% CI: 1.705, 4.918, *p* = 0.000, Figure 7) as compared to the control group, while FM (WMD = −2.221; 95% CI: −4.582, 0.139, *p* = 0.065, Figure 8) and FFM (WMD = 0.487; 95% CI: −3.512, 4.469, *p* = 0.814, Figure 9) remained unchanged. No significant heterogeneity was detected in all indicators.

## 4. Discussion

This meta-analysis evaluated the impact of LCHF on human body composition with the addition of HIIT. The results of the ten studies included for analysis show that LCHF significantly decreased human BM, BMI, and BFP by the combination with HIIT, while FM and FFM remained unchanged. Moreover, VO_2max_ was augmented following the intervention.

Recently, attention has been paid to LCHF on account of the reported influences on body composition. Extensive research has demonstrated the remarkable effect of LCHF on weight loss and fat loss [9,10,11] [9 (RCT), 10 (RCT), 11 (RCT)]. Similar to the previous studies, our results support the evidence that individuals that adhere to LCHF diet exhibit a larger decline in BM, FM, and BFP than those assigned to a non-LCHF diet group. The underlying mechanisms of LCHF-induced weight and fat reduction may relate to a limited appetite or the changing metabolic modulators. Some investigations indicated that the LCHF diet elicits physiological ketosis, which is characterized by the increment in ketone bodies, such as β-hydroxybutyrate, acetoacetate, and acetone [56,59]. The ketosis effect is assumed to suppress appetite [57] and thus decrease total calorie intake, as multiple kinds of research find the LCHF diet group consumes fewer calories in ad libitum patterns [17,19,20,60]. More importantly, this view is confirmed by the fact that the LCHF diet group only resulted in more BM and FM loss in ad libitum studies [17,19,20,60] but not in isocaloric studies [18,58,61]. However, on the other hand, Sun et al. performed a four-week isocaloric LCHF dietary pattern and still detected a reduction in BM and BMI in overweight/obese Chinese females [43], which suggests this weight-reducing effect may be generated by other possible reasons. Lipogenesis reduction and lipolysis escalation may be the plausible mechanisms of weight and fat loss for the isocaloric LCHF diet, as proven by the decreased insulin [62]. Another possible reason may involve the extra need for gluconeogenesis for energy production, which is a process consuming extra energy [63,64]. Nevertheless, dietary intervention alone may bring a negative impact on muscle mass. Therefore, the combination with an exercise intervention, such as HIIT and RT, has received lots of attention. A recent meta-analysis indicated that the ketogenic diet decreased FFM, and this negative effect could not be ameliorated by combining it with RT [36]. However, surprisingly, our result reveals that LCHF combined with HIIT has no significant effect on FFM, which implies that HIIT prevents greater muscle mass loss during LCHF than RT.

Except for the weight-reducing effect, the LCHF diet may also become a possible dietary intervention to enhance sports performance. The LCHF diet has been demonstrated to cause a decline in total carbohydrate oxidation and an increase in fat oxidation and lipolysis during prolonged exercise [63,65], which would in turn improve exercise performance. However, there is much debate on its effect on VO_2max_, as some studies found a reduction in the VO_2max_ of the LCHF diet group [42], whereas others did not find this [16,43]. A systematic review proposed that the duration of consuming LCHF may be an essential factor affecting physical performance. Long-term intervention demands metabolic adaptations to minimize adverse effects [37]. Our meta-analysis indicated that LCHF combined with HIIT for 2–14 weeks has a positive effect on VO_2max_, which suggests that the negative effect at the initiation of the diet can be reversed by the addition of HIIT.

The scope of this study was limited by the following terms. First, since all included studies lasted less than 14 weeks, we are unable to explain the long-term effect of LCHF combined with HIIT on body composition and VO_2max_. Second, the included studies employed different estimates of VO_2max_ (i.e., VO_2max_ or VO_2peak_), which may lead to deviation. Third, dietary intervention and modes of HIIT were performed in different patterns (i.e., ad libitum or isocaloric, cycling or unarmed training). Fourth, the populations involved were of different types (i.e., overweight/obese people, healthy untrained individuals, or elite athletes).

## 5. Conclusions

LCHF combined with HIIT reduces body weight and fat mass while maintaining lean body mass and enhancing aerobic capacity. However, considering the adverse effects of LCHF, whether its combination with HIIT can be used in a range of patients and its long-term safety remains unknown. Further large, long-term, well-designed randomized trials on this topic are needed to assess the long-term safety, efficacy, and adherence to the combination of LCHF and HIIT. Further studies are also necessary to explore the impact of other types of exercise (i.e., moderate-intensity interval training, aerobic training, etc.) on body composition and physical performance during LCHF.

## Figures and Tables

**Figure 1 ijerph-18-10740-f001:**
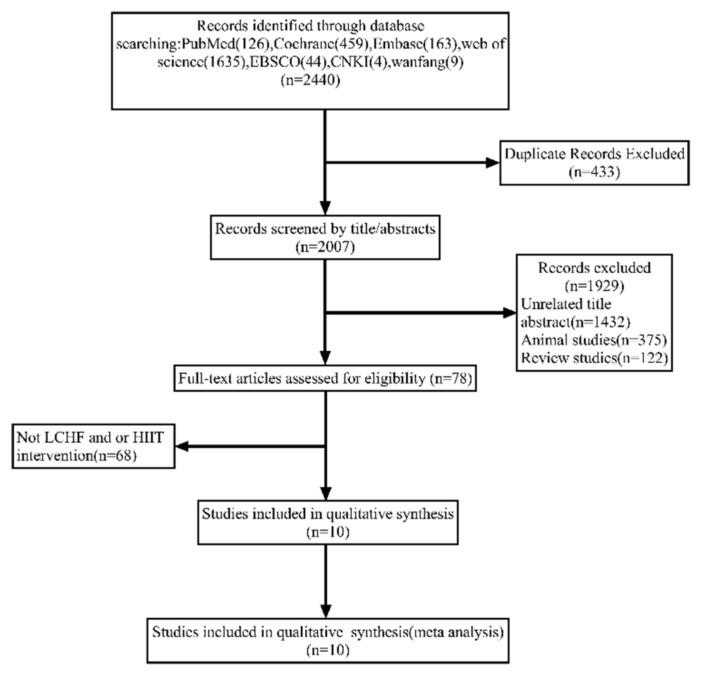
Flow diagram of the study selection process.

**Figure 2 ijerph-18-10740-f002:**
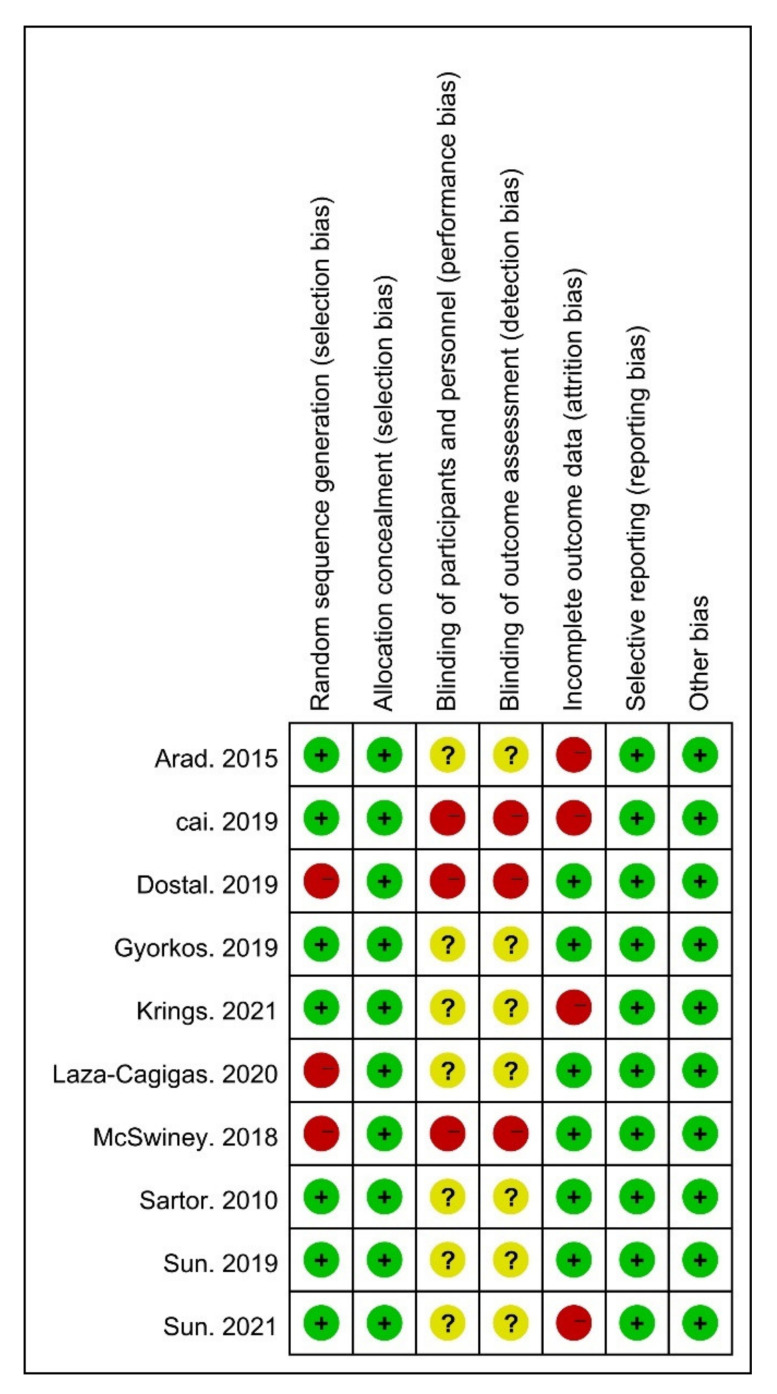
Risk of bias (ROB) results of quality assessment within included studies. Risk of bias levels: low (green or “+”), unclear (yellow or “?”), and high (red or “–”).

**Figure 3 ijerph-18-10740-f003:**
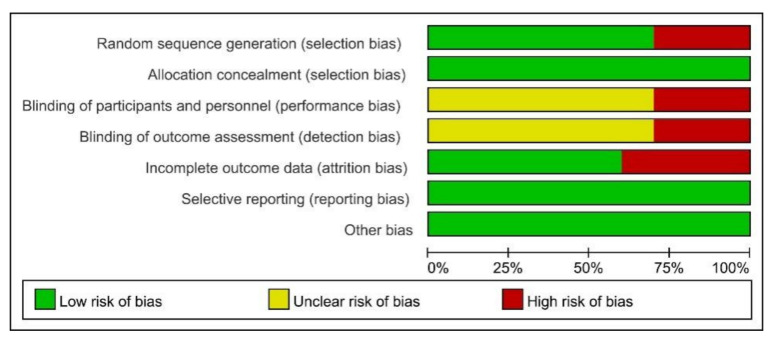
Risk of bias (ROB) results of quality assessment within included studies. Risk of bias levels: low (green), unclear (yellow), and high (red).

**Figure 4 ijerph-18-10740-f004:**
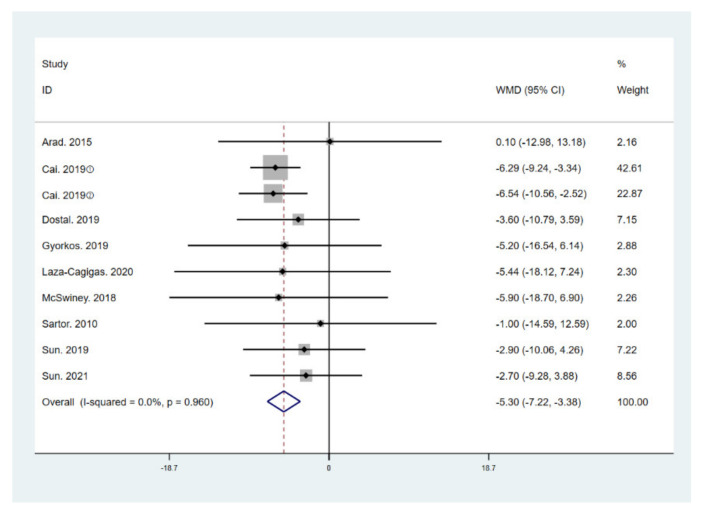
Forest plot for the effect of a low-carbohydrate high-fat (LCHF) diet combined with HIIT on body mass (BM). For each study, squares represent the mean difference in intervention effects, with horizontal lines intersecting them as the lower and upper limits of the 95% CI. The size of each square represents the relative weight of the studies conducted in the meta-analysis. The diamond represents the results of the meta-analysis combining the individual studies.

**Figure 5 ijerph-18-10740-f005:**
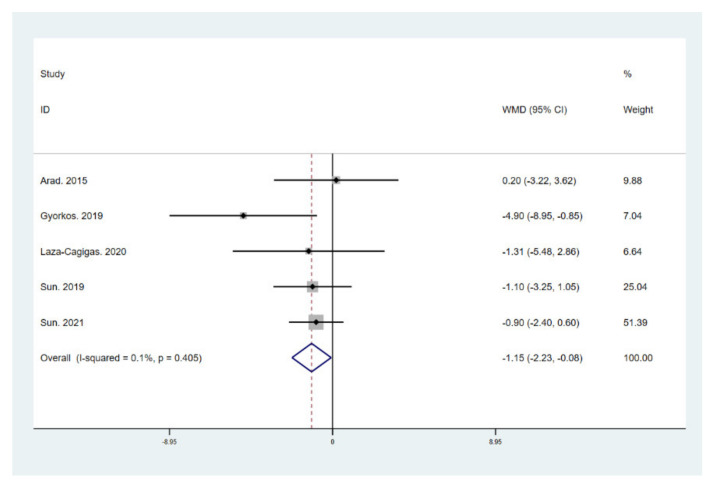
Forest plot for the effect of a low-carbohydrate high-fat (LCHF) diet combined with HIIT on body mass index (BMI). For each study, squares represent the mean difference in intervention effects, with horizontal lines intersecting them as the lower and upper limits of the 95% CI. The size of each square represents the relative weight of the studies conducted in the meta-analysis. The diamond represents the results of the meta-analysis combining the individual studies.

**Figure 6 ijerph-18-10740-f006:**
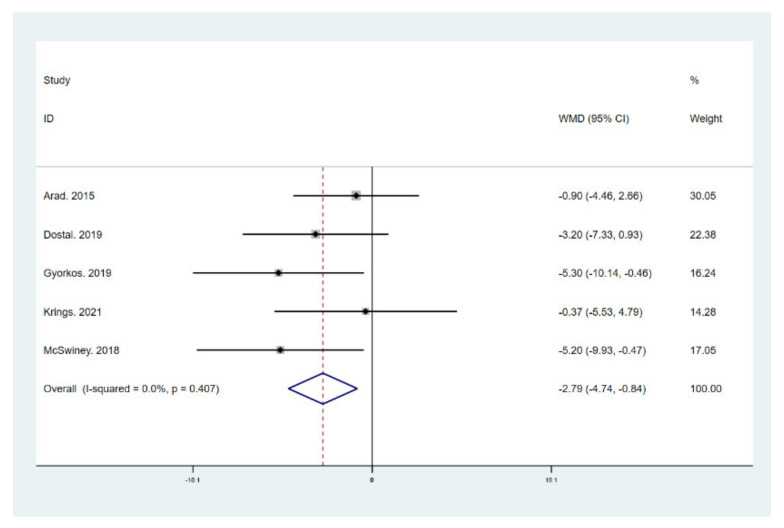
Forest plot for the effect of a low-carbohydrate high-fat (LCHF) diet combined with HIIT on body fat percentage (BFP). For each study, squares represent the mean difference in intervention effects, with horizontal lines intersecting them as the lower and upper limits of the 95% CI. The size of each square represents the relative weight of the studies conducted in the meta-analysis. The diamond represents the results of the meta-analysis combining the individual studies.

**Figure 7 ijerph-18-10740-f007:**
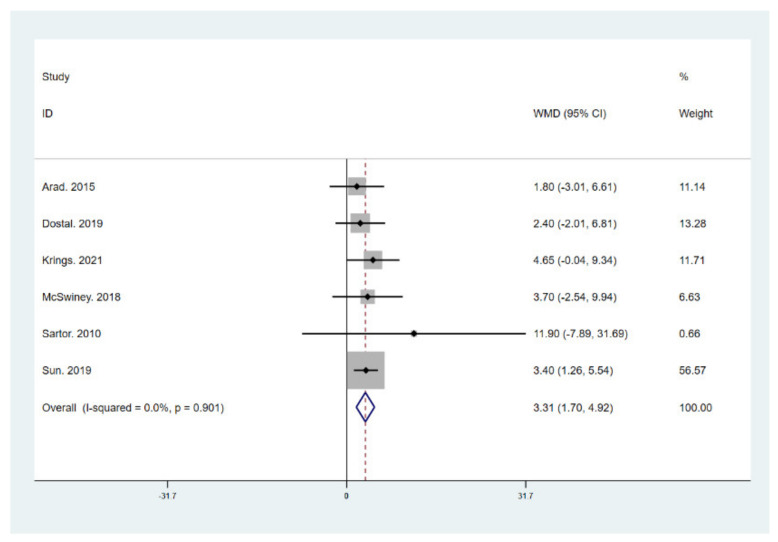
Forest plot for the effect of a low-carbohydrate high-fat (LCHF) diet combined with HIIT on maximal oxygen uptake (VO_2_ max). For each study, squares represent the mean difference in intervention effects, with horizontal lines intersecting them as the lower and upper limits of the 95% CI. The size of each square represents the relative weight of the studies conducted in the meta-analysis. The diamond represents the results of the meta-analysis combining the individual studies.

**Figure 8 ijerph-18-10740-f008:**
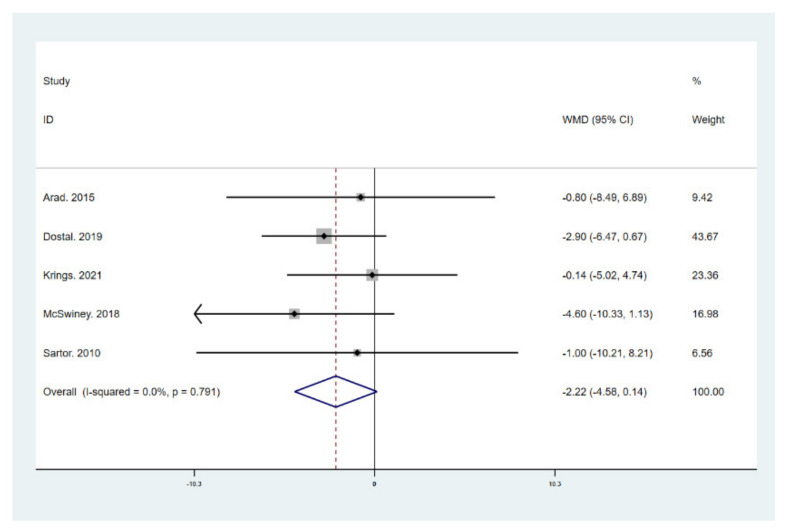
Forest plot for the effect of a low-carbohydrate high-fat (LCHF) diet combined with HIIT on fat mass (FM). For each study, squares represent the mean difference in intervention effects, with horizontal lines intersecting them as the lower and upper limits of the 95% CI. The size of each square represents the relative weight of the studies conducted in the meta-analysis. The diamond represents the results of the meta-analysis combining the individual studies.

**Figure 9 ijerph-18-10740-f009:**
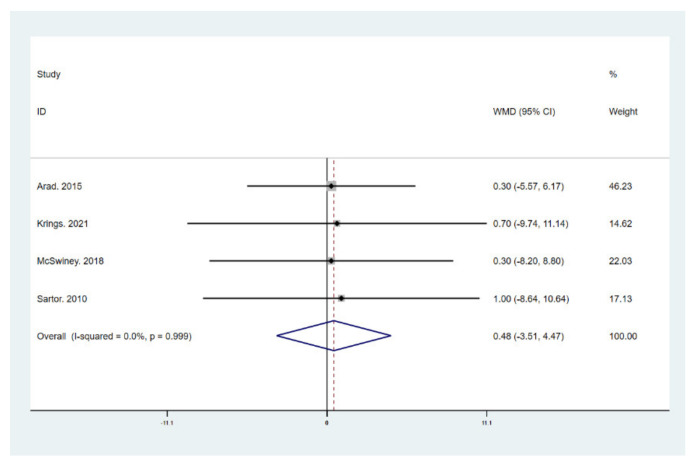
Forest plot for the effect of a low-carbohydrate high-fat (LCHF) diet combined with HIIT on fat free mass (FFM). For each study, squares represent the mean difference in intervention effects, with horizontal lines intersecting them as the lower and upper limits of the 95% CI. The size of each square represents the relative weight of the studies conducted in the meta-analysis. The diamond represents the results of the meta-analysis combining the individual studies.

**Table 1 ijerph-18-10740-t001:** Characteristics of studies included in the meta-analysis.

Study	Country	Type of Study	Participants Characteristic/Age/N (Male/Female)	HIIT Intervention/Duration	Macronutrient Ratio in LCHF	Type of Diet	Macronutrient Ratio in Non-LCHF	Result of BM and BMI	Result of FM and BFP	Result of FFM	Result of VO_2max_
Arad et al. 2015 [50]	United States	RCT	Overweight/obese females 20–40 years 9 (0/9)	4 bouts of 30–60 s high-intensity cycling followed by 180–210 s recovery (24 min in total) of 3 sessions/week 14 weeks	CHO: 35% Prt: 15% Fat: 50% Energy: resting metabolic rate ×1.5	LCHF	Not mentioned	↔	↔	↔	↔
Cai et al. 2019 ① [54]	China	RCT	Overweight/obese males and females 29.57 ± 3.69 years 7 (1/6)	1 min HIIT training followed by 20 s recovery (6 times in total) of 2 sessions/week 1 month	CHO: 30 g Prt: 60 g Fat: 130 g Energy: 1500 ± 50 kcal	KD	CHO: 250 g Prt: 60 g Fat: 20 g Energy: 1500 ± 50 kcal	↓	-	-	-
Cai et al. 2019 ② [54]	China	RCT	Overweight/obese males and females 29.40 ± 3.06 years 10 (2/8)	1 min HIIT training followed by 20 s recovery (6 times in total) of 2 sessions/week 1 month	CHO: 30 g Prt: 60 g Fat: 130 g Energy: 1500 ± 50 kcal	KD	CHO: 250 g Prt: 60 g Fat: 20 g Energy: 1500 ± 50 kcal	↓	-	-	-
Dostal et al. 2019 [41]	The Czech Republic	Non-RCT	Healthy males and females 18–35 years 12 (3/9)	5 × 6 min sets separated by 2 min recovery (40 min in total) of 3–5 sessions/week 12 weeks	CHO: 40 ± 6 g Prt: 113 ± 24 g Fat: 149 ± 26 g Energy: 8206 ± 1322 kJ	VLCHF	CHO: 194 ± 43 g Prt: 79 ± 21 g Fat: 71 ± 17 g Energy: 7523 ± 1541 kJ	↔	↔	-	↑
Gyorkos et al. 2019 [52]	United States	RCT	Metabolic syndrome males and females 18–60 years 12 (4/9)	60 s cycling intervals interspersed with 60 s of active recovery, and a three min cool-down (10 times in total) of 3 sessions/week 4 weeks	CHO: 51 ± 7 g Prt: 87 ± 30 g Fat: 118 ± 27 g Energy: 1590 ± 587 kcal	CRPD	CHO: 277 ± 105 g Prt: 80 ± 36 g Fat: 117 ± 31 g Energy: 2466 ± 602 kcal	↓	↓	-	↑
Krings et al. 2021 [48]	China	RCT	Healthy males 19.7 ± 1.0 years 9 (9/0)	30 s maximal sprints interspersed with 4.5 min of active recovery (4–8 times in total) of 2 sessions/week 4 weeks	CHO: 25% Prt: 25% Fat: 50% Energy: 3051.30 ± 504.02 kcal	CRD	CHO: 298.71 ± 51.03 g Prt: 124.86 ± 35.60 g Fat: 101.14 ± 16.12 g Energy: 2637.92 ± 390.61 kcal	-	↔	↔	↑
Laza-Cagigas et al. 2020 [55]	United Kingdom	Non-RCT	Type 1 or Type 2 diabetes 63–79 years 16 (13/3)	180 s HIIT training followed by 120 slow-intensity intervals (4 times in total) of 2 sessions/week less than 9 weeks	Not mention	LCHF	Not mention	↓	-	-	-
McSwiney et al. 2018 [49]	Ireland	Non-RCT	Endurance-trained male athletes 18–40 years 9 (9/0)	10 sets of 1 min bouts at 70% peak power with 1 min recovery of 2 sessions/week 12 weeks	CHO: 41.1 ± 13.3 g Prt: 130.7 ± 35.8 g Fat: 259.3 ± 83.4 g Energy: 3022.3 ± 911.1 kcal	LCKD	CHO: 454.8 ± 152.0 g Prt: 110.3 ± 25.5 g Fat: 64.7 ± 39.1 g Energy: 2843.8 ± 558.4 kcal	↓	↓	↔	↔
Sartor et al. 2010 [53]	Greece	RCT	Obese males and females 37 ± 10 years 10 (2/8)	4 min bouts at 90% VO_2_ peak with 2–3 min rest on cycle ergometers (up to 10 times) of 3 sessions/week 2 weeks	CHO: 163 ± 30 g Prt: 70.9 ± 9.8 g Fat: 63.8 ± 12.6 g (unsaturated fat), 33.5 ± 8.0 g (saturated fat) Energy: 1886 ± 345 kcal	CRD	CHO: 304 ± 57 g Prt: 91.7 ± 26.2 g Fat: 44.9 ± 17.5 g (unsaturated fat) 35.7 ± 11.8 g (saturated fat) Energy: 2363 ± 452 kcal	↓	↓	↑	↑
Sun et al. 2019 [43]	China	RCT	Overweight/obese females 18–30 years 18 (0/18)	10 bouts of 6 s cycling sprints followed by 9 s passive recovery (2.5 min in total) of 5 sessions/week 4 weeks	CHO: 49 ± 17 g Prt: 109 ± 28 g Fat: 137 ± 17 g Energy: 1871 ± 246 kcal	LC	CHO: 236 ± 59 g Prt: 77 ± 23 g Fat: 84 ± 25 g Energy: 2057 ± 437 kcal	↔	-	-	↔
Sun et al. 2021 [51]	China	RCT	Overweight/obese females 21.4 ± 2.9 years 17 (0/17)	10 repetitions of 6 s cycling sprints interspersed with 9 s passive recovery (2.5 min in total) of 5 sessions/week 4 weeks	CHO: 46 ± 15 g Prt: 111 ± 25 g Fat: 141 ± 18 g Energy: 1828 ± 204 kcal	LC	CHO: 241 ± 58 g Prt: 79 ± 19 g Fat: 89 ± 22 g Energy: 2071 ± 407 kcal	↓	-	-	-

LCHF: low-carbohydrate high-fat, KD: ketogenic diet, VLCHF: very-low-carbohydrate high-fat, CRPD: carbohydrate-restricted paleolithic-based diet, CRD: carbohydrate-restricted diet, LCKD: low-carbohydrate high-fat ketogenic diet, LC: low-carbohydrate diet.

## Data Availability

All the included studies are in Table 1.

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
