# Peer review of "Effects of a Low-Carbohydrate High-Fat Diet Combined with High-Intensity Interval Training on Body Composition and Maximal Oxygen Uptake: A Systematic Review and Meta-Analysis"

_ijerph, 2021, doi:10.3390/ijerph182010740_

Round 1

Reviewer 1 Report

Dear Authors,

This review is dealing with the effects of LCHF combined with HITT on body composition and maximal oxygen uptake. The paper is interesting and addresses an important topic. However, in order to make the paper publishable, the following points should be addressed.

 Introduction

  • In general, this section is quite short. It is requested that it be increased considerably, and if possible that more references are included.
  • Line 33-36: More details are needed. What are the different types of LCHF diets? The ketogenic diet (KD) and Atkins diet are some examples.
  • It is not clear if this review focuses on KD or other LCHF types. Why is KD better than other LCHF diets? Is a KD recommendable?
  • Lines 36,40,44,64,193: Please define the design of these studies (i.e cohort, RCT).
  • Lines 37,57,63 (LCHF diet): Are you referring to KD here?
  • Line 39: Some literature states the effectiveness of KD in managing obesity. How about recent literature? What says about the role of KD in treating obesity and related diseases in general population? Authors should state that obesity is a complex disease which increased risk of cardiovascular diseases (CVD), and leads to a state of low-grade systemic inflammation that may cause chronic systemic inflammatory disorders such as asthma. Is there new evidence to suggest that the KD could have a protective role against obesity, CVD and asthma in general population? I would suggest authors referring to the following reviews: Can Fam Physician. 2018, 64, 906; Int. J. Mol. Sci. 2020, 21(24), 9580; Nutrients. 2017, 9, 517).
  • Line 43-44: Please expand on these beneficial changes in body composition, and how HIIT or RT increase LM and FFM.
  • Line 67-69: The novelty aspect of the review could be improved. What were your research hypotheses?

Methods  

  • Line 77-86: Many of the search terms reported are unclear and repetitive. For clarity, please rephrase.
  • Line 89-94: In terms of research quality, deciding on inclusion and exclusion criteria is not clearly defined. What age group are adults? What is their gender? What types of dietary/exercise intervention used? Are they RCTs? What types of LCHF diets used? I understand these details are found in results, but should also include here.
  • Line 96-97: A reference should be included to support this statement.

Results

  • Line 122-123: 68 articles were excluded due to not relevant information- meaning unclear. Please clarify both in text and figure.
  • Line 127: Studie should be studies.
  • Table 1 is well-designed. Can you please clarify if it is focus on KD or other LCHF types? I suggest adding a column defining a country where the study was conduced.
  • Figures 4-9 are not readable, please clarify.

Conclusions

  • Line 237-242: This section is fine but needs to be strengthened and expanded.

Author Response

Dear reviewers and editors,

We are sorry for our oversight. The comments have been carefully taken into account, and a newly revised submission has been uploaded. We highlighted all the altered passages in red. Replies are in the attachment. Special thanks to you for your good comments.

Reviewer 2 Report

This study systematic review and meta-analysis examined the effect of low carbohydrate high fat diet (LCHF) combine with high-intensity interval training (HIIT) on body composition and maximal oxygen uptake. The findings show that LCHF diet combined with HIIT can reduce weight and fat effectively. This combination is sufficient to prevent muscle mass loss during LCHF, and further enhance maximal oxygen uptake.

Overall, the manuscript is properly structured and well written. I have a few concerns.

  1. The manuscript only included one study comprising athletes (McSwiney et al., 2018), who are largely different from the healthy, overweight/obese and clinical populations. In fact, if the trained population was included in this review and meta-analysis, it seems that the authors may omit some relevant studies (https://doi.org/10.3390/nu13082896; https://doi.org/10.1186/s12970-020-00362-9).
  2. My suggestion is no need to include the studies of non-trained individuals based on the focus of the present study and the fact of some systematic reviews and meta-analyses have been previously published.
  3. Line 103 of Page 3, please report the name of the third researcher.

Author Response

(The authors gave the same response as above.)

Round 2

Reviewer 1 Report

No further comments.